# P-Tracker: Design and Development of a Low-Cost PM2.5 Monitor for Citizen Measurements of Air Pollution

**Marks Jalisevs** [1,*], **Hamza Qadeer** [2], **David O'Connor** [3], **Mingming Liu** [1,4] **and Shirley M. Coyle** [1,4,*]

1   School of Electronic Engineering, Dublin City University, D09 DD7R Glasnevin, Ireland; mingming.liu@dcu.ie
2   School of Computing, Dublin City University, D09 K2WA Glasnevin, Ireland; hamza.qadeer@dcu.ie
3   School of Chemical Sciences, Dublin City University, D09 K20V Glasnevin, Ireland; david.x.oconnor@dcu.ie
4   Insight Research Ireland Centre for Data Analytics, Dublin City University, D09 YH9P Glasnevin, Ireland
*   Correspondence: marks.jalisevs@gmail.com (M.J.); shirley.coyle@dcu.ie (S.M.C.)

**Abstract**

Particulate matter (PM2.5) is a critical indicator of air quality and has significant health implications. This study presents the development and evaluation of a custom-built PM2.5 device, named the P-Tracker, designed to offer an accessible alternative to commercially available air quality monitors. This paper presents the design framework used to address the requirements of a low-cost, accessible device which meets the performance of existing commercial systems. Step-by step build instructions are provided for hardware and software development and connection to the P-tracker open access website which displays the data and interactive map. To demonstrate the performance, the P-Tracker was compared against leading consumer devices, including the AtmoTube Pro by AtmoTech Inc., Flow by Plume Labs, View Plus by Airthings, and the Smart Citizen Kit 2.1 by Fab Lab Barcelona, across four controlled tests. The tests included: (1) a controlled paper combustion test in which all devices were exposed to combustion aerosols in a sealed environment alongside the DustTrak 8530 (TSI Incorporated, Shoreview, MN, USA), used as the gold standard reference, where the P-Tracker achieved a Pearson correlation of 0.99 with DustTrak over the final measurement period; (2) an outdoor test comparing readings with a stationary reference sensor, Osiris (Turnkey Instruments Ltd., Rudheath, UK), where the P-Tracker recorded a mean PM2.5 concentration of 3.08 μg/m$^3$, closely aligning with the Osiris measurement of 3.53 μg/m$^3$ and achieving a Pearson correlation of 0.77; (3) a controlled indoor air quality assessment, where the P-Tracker displayed stable readings with a standard deviation of 0.11 μg/m$^3$, comparable to the AtmoTube Pro; and (4) a real-world kitchen environment test, where the P-Tracker effectively captured fluctuations in PM2.5 levels due to cooking activities, maintaining a consistent response with the DustTrak reference. The results indicate varied degrees of agreement across devices in different conditions, with the P-Tracker demonstrating strong correlation and low error margins in high-pollution and controlled scenarios. This research underscores the potential of open-source, low-cost, custom-built air quality sensors which may be developed and deployed by communities to provide hyperlocal measurements of air pollution.

**Keywords:** particulate matter; PM2.5; air quality sensors; air pollution; citizen science; environmental health

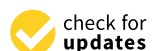

## 1. Introduction

Air pollution poses significant risks to public health, contributing to respiratory and cardiovascular diseases, particularly in urban environments where pollutant levels can

vary widely over short distances and time scales. Fine particulate matter, specifically PM2.5 (particles with a diameter of less than 2.5 μm), is of particular concern due to its ability to penetrate deep into the respiratory system, causing significant health challenges [1,2]. Monitoring PM2.5 concentrations is crucial for understanding exposure patterns, mitigation of risk, and informing public health policies. However, traditional air quality monitoring stations, while highly accurate, are typically expensive, and stationary (fixed locations). This can lead to a limited distribution, which can fail to capture the high variability of air pollution across communities, leaving significant gaps in understanding personal exposure levels, especially for individuals who move through diverse environments daily. The aim of this work is to support a citizen science approach to air quality measurements, providing open-source solutions to prototyping for the research community and those interested in creating their own air quality monitoring systems to understand personal exposure to pollution.

### 1.1. Particulate Matter Introduction

Particulate Matter (PM) refers to tiny particles that can be either solid or liquid form. While some of these particles are naturally occurring, many are produced by human activities [3]. There are different sizes of particles that contribute to air pollution, two commonly used terms are PM2.5 and PM10, where PM10 means that the particulates are 10 microns and smaller, and for PM2.5 the particles are 2.5 microns and smaller. The PM10 particles are about 1/10th the width of an average human hair, meaning you could stack about ten PM10 particles to equal the thickness of a single human hair. In contrast, PM2.5 particles are much smaller, requiring around 40, PM2.5 particles to equal the thickness of a human hair [3]. The PM2.5 is a mixture of the different emissions and chemicals, that are emitted from vehicles, gasoline and diesel fuel, industrial emission as factories and power plants, organic ions such as sulfur dioxide and nitrogen oxide and wildfires [4]. The PM10 comes from the same mixture of different chemical compositions but includes larger particles such as dust, pollen, mold spores, and other materials from mechanical processes like construction activities, road dust, agricultural operations, and natural sources like soil and sea salt aerosols [4]. Both PM2.5 and PM10 in the air can cause serious health issues when inhaled. PM2.5 can penetrate deep into the lungs and even enter the bloodstream, leading to conditions like heart attacks, asthma, and decreased lung function. The PM10 is larger but can still cause respiratory problems. Both types of particles are especially harmful to vulnerable groups, including people with heart or lung disease, children, and the elderly. These particles also contribute to environmental problems like reduced visibility and damage to ecosystems [5].

### 1.2. Advances in Citizen-Based Air Quality Sensing

To address these limitations, the development of low-cost, portable air quality devices has gained significant attention in recent years. However, existing solutions still face challenges that limit their effectiveness for real-time, mobile, and independent air quality monitoring. There is growing interest in this research area, while earlier devices such as the works presented in [6,7] had limited long-term functionality due to lack of portability and wireless connectivity, and also required external power and manual retrieval of stored data. Dynamic, real-world applications require real-time data transmission and power autonomy. The commercial Smart Citizen Kit [8] also presents limitations, particularly its lack of GPS functionality and its stationary design. While useful for long-term environmental monitoring in a fixed location, it is not suitable for mobile air quality tracking. Without GPS, the device cannot correlate pollution levels with specific locations, limiting its use in studies that require geospatial air quality mapping. The device described in [9] shares

many of the same limitations as those in [6,7], including no wireless data transmission and no built-in battery, and also lacks GPS integration. This means that while it can record air quality data, it does not provide spatial context, making it less effective for applications that require mobile monitoring and location-based pollution analysis. The SnifferBike system in [10,11] introduces a different approach by integrating air quality monitoring with bicycles. However, it depends on mobile network. Additionally, it has no offline data storage, meaning data loss is possible if real-time transmission fails. Its limited battery life restricts long-term use, and because it is designed specifically for bicycles, it may not be suitable for handheld use, reducing its flexibility for broader applications.

PurpleAir sensors [12] are another example of low-cost devices which are designed for community driven air quality monitors, suitable for both indoor and outdoor use. Research studies and community initiatives have investigated the effects of wildfire smoke and fireworks displays along with ventilation within indoor settings. The device has a dual PMS6003 particulate matter sensor for redundancy and improved reliability and provides real-time PM2.5 readings via Wi-Fi to a publicly available online map. The device includes optional offline storage with a MicroSD card but does not have GPS functionality for portability function.

The P-Tracker was designed to overcome the challenges identified in existing air quality monitoring devices, and provide citizens and researchers with the tools to build their own sensing device. Unlike previous systems [6,7], which lack Wi-Fi/Bluetooth connectivity and a built-in battery, the P-Tracker operates independently with a rechargeable power source, allowing for extended portable use. Compared to stationary devices like Smart Citizen Kit [8] and other research-based monitors [9], the P-Tracker integrates GPS tracking, enabling precise location tagging of air quality measurements, making it ideal for mobile pollution mapping. Additionally, it offers offline data storage via an SD card, unlike SnifferBike, ensuring continuous data collection even in areas without network access. To provide a practical and user-friendly solution, the P-Tracker integrates key components such as the ESP32 microcontroller and SDS011 particulate matter sensor, ensuring PM2.5 and PM10 readings. The GPS module records geolocation data, allowing users to visualize pollution exposure on an interactive map via a web-based interface. This eliminates the need for proprietary mobile apps and enhances accessibility for real-time and historical data analysis. Additionally, the device includes a function to calculate average pollution exposure over a user's journey, offering deeper insights into personal air quality levels.

### 1.3. Comparing the P-Tracker with Commercially Available Devices

To evaluate the efficiency of the P-Tracker, its performance was compared with commercially available portable and stationary air quality devices, including the AtmoTube Pro (by Atmotech Inc., San Francisco, CA, USA), Flow (by Plume Labs, Paris, France), View Plus (by Airthings, Oslo, Norway), and the Smart Citizen Kit 2.1 (by Fab Lab Barcelona, Barcelona, Spain). A series of tests were conducted to assess the accuracy and reliability of these devices under various conditions. These included (1) a controlled paper combustion test in a sealed environment with DustTrak 8530 as a reference device, (2) an outdoor test comparing readings with the stationary Osiris sensor (by Turnkey Instruments Ltd., Rudheath, UK), (3) indoor air quality monitoring, and (4) a kitchen environment test to simulate real-world conditions. This comprehensive evaluation highlights the potential of low-cost portable devices for air quality monitoring and their role in supporting individuals and communities to make informed decisions based on environmental data.

The findings from this paper contribute to the growing field of research on portable air quality devices and sensors, emphasizing their utility in bridging the gap between personal exposure monitoring and traditional air quality networks. By enabling real-time,

location-specific measurements, devices like the P-Tracker represent a step forward in the democratization of air quality data, promoting greater public awareness and engagement in environmental health.

## 2. Design

### 2.1. P-Tracker's Air Quality Sensor SDS011

The SDS011 sensor, developed by Nova Fitness Co., Ltd., is a highly reliable and precise particulate matter sensor, designed for monitoring air quality by measuring concentrations of PM2.5 and PM10 [13]. Utilizing the principle of laser scattering, the SDS011 can detect particles ranging from 0.3 to 10 μm, making it particularly effective in identifying fine particles that pose significant health risks. The sensor features a digital output and a built-in fan that ensures consistent airflow, thereby enhancing measurement accuracy and response time, which is less than 10 s when there is a change in the surrounding environment. The SDS011 outputs data via UART, allowing for easy connection with microcontroller and other data processing units. With a high resolution of 0.3 $\mu g/m^3$, this sensor is suitable for applications such as air quality monitoring, purifiers, and personal exposure assessments. The sensor is also certified under CE, FCC, and RoHS standards, ensuring its compliance with international safety and environmental regulations [13].

### 2.2. Calibration Methodology

The P-tracker uses the SDS011 for PM measurements which is supplied with factory calibration from the manufacturer, Nova Fitness Co., Ltd. The factory calibration adjusts each sensor during the production process. In this study, no extra calibration procedure was applied. To ensure reliability and performance of the SD011, the sensor's readings were compared with the reference grade devices, DustTrak and Osiris.

### 2.3. Sensor Limitations and Maintanance

One of the most important limitations of the SDS011 is the sensitivity to the high relative humidity. The laser scattering technology which is used in the SDS011 sensor is affected by hygroscopic growth of particles and fog or condensed water droplets which can lead to significant overestimation of the particulate matter concentration. A study by Han et al. shows how sensors utilizing laser scattering detection techniques are affected by relative humidity, condensation and fog. When relative humidity exceeds 70% the laser scattering sensors are reported to record unreliable readings, which is more prevalent in the early morning and evening periods. The error in the particulate matter readings was found to mostly occur during foggy conditions, where the sensor detects fog droplets as PM-sized particles and artificially increments the measured concentration [14].

Another study by Budde et al. found humidity to be a critical limitation for the SD011, especially in foggy conditions. This occurs due to the hygroscopic growth of particles, which become larger and scatter more light resulting in the rise in particulate matter concentration. Also, SDS011 struggles with accurately detecting particles of various sizes. A controlled laboratory experiment using the monodisperse polystyrene particles shows that the sensor tends to underestimate concentrations of larger particles and overestimate or inaccurately measure the PM2.5 concentration in specific ranges of particle size [15].

Despite the advanced functionality of the SDS011, the sensor's lifespan is limited by the laser diode, which can last up to 8000 h under continuous use. Assuming the device is used for 10 h daily, this would translate to 800 days or 2.19 years of operation. At the current price of €24.81, the annual cost of the sensor is approximately €11.33 per year. With continuous use, the performance may degrade because of the fan wear or collection of internal dust. These issues can affect the detection of the particulate matter

and alter measurements. This aging effect is a challenge in most low-cost sensors. To ensure performance over an extended period for the SDS011, the inlet and outlet openings need to be cleaned to prevent dust accumulation and to help maintain proper airflow. As the SDS011 operates using laser scattering technology, the built-up dust inside the sensor can affect the measurements' quality. The sensors fan is a small component with limited lifespan also. Monitoring fan noise or irregular air flow could help to detect potential issues with the fan. Another approach to extend the SDS011 working lifespan would be to configure the sensor to operate periodically with sleep modes, for example, waking up the sensor once every 10–15 min to measure particulate matter and then returning to sleep mode again. Nevertheless, the low operation cost makes the SDS011 an economical option for long-term low-cost air quality devices.

*2.4. Hardware Description and Rationale*

The device, named P-Tracker, is designed as a low-cost, portable air quality monitoring tool for measuring PM2.5 and PM10 concentrations and temperature and humidity. The following components were selected for their performance, compatibility, low power consumption and affordability:

- ESP32 Microcontroller: Chosen for its dual-core processing power, built-in Wi-Fi and Bluetooth, and compact form factor, providing a balance of performance and energy efficiency [16].
- SDS011 Sensor: A laser scattering-based particulate matter sensor was selected for its accuracy in measuring PM2.5 and PM10, quick response time, and cost-effectiveness [13].
- GPS Module (NEO M6): Enables precise geolocation tracking to correlate pollution data with specific locations. Its low power consumption and high sensitivity make it suitable for mobile applications [17].
- DS3231 RTC Module: Provides time and temperature data with high precision, essential for timestamping air quality measurements [18].
- 0.96-inch OLED Display: Used for real-time visualization of PM2.5 and PM10 levels, Wi-Fi connection status, temperature, humidity and time. Its high contrast enhances readability in various environments [19].
- MicroSD Card Reader: Ensures data are saved locally as a backup, especially in areas with unreliable Wi-Fi connectivity [20].
- $3 \times 2000$ m/Ah 18650 Lithium-Ion Batteries: Chosen for their high energy density and rechargeability, supporting up to 40 h of operation under typical conditions. Managed by a DollaTek 18650-Board for efficient charging and power distribution [21].
- HTU21D Temperature and humidity sensor: low-cost and low power consumption sensor [22].

*2.5. Design Framework*

The P-Tracker emphasizes portability, modularity, and energy efficiency:

- Portability: The enclosure was compactly designed using recycled materials, allowing the device to be carried or attached to bicycles or electric scooters.
- Energy Efficiency: Efficient operation of the SDS011 and other modules and sensors extend its working period and minimizes battery drain.
- Cost-Effectiveness: At an approximate total cost of €40 for the components, the P-Tracker is an accessible solution, significantly more affordable compared to commercial alternatives like the AtmoTube Pro €189 [23], the Flow by Plume Labs €180 [24], and the Smart Citizen Kit 2.1 €171 [25].

*2.6. Advantages and Disadvantages Compared to Alternatives*

Advantages:

- Low cost makes it accessible for citizen science applications.
- Offers real-time data and GPS-based geotagging.
- Recyclable materials in the enclosure align with sustainable practices.
- Open-source code provided on Arduino IDE, available for all to access.
- Low power consumption, with a total battery capacity of 6000 m/Ah.
- Portable device, suitable for mobile recording of air quality measurements.

Disadvantages:

- Limited lifespan of the SDS011 sensor (up to 8000 h).
- Enclosure size needs sufficient space, and the device may be too heavy for wearable applications.
- Device must be securely fastened as the prototype device may be fragile if dropped.
- SDS011 is sensitive to humidity, necessitating additional humidity measurements and correction algorithms.

*2.7. Component Overview and Cost*

Table 1 below shows the prices for the components that were used to build the P-tracker. The prices given are approximate due to price fluctuations depending on the time of purchase. Miscellaneous parts such as switches, small screws and wires are not listed as those are sold only in bulk and their cost is minimal.

**Table 1.** Bill of Materials.

| Quantity | Component | Source of Materials | Material Type | Cost |
|:---:|:---:|:---:|:---:|:---:|
| 1 | ESP32 (by DOIT, Kunshan, China) | AliExpress | Microcontroller | €3.03 |
| 1 | SDS011 (by Nova Fitness Co., Ltd., Jinan, China) | AliExpress | PM2.5 and PM10 sensor | €24.82 |
| 1 | NEO M6 GPS (by u-blox, Thalwil, Switzerland) | AliExpress | Location module | €3.18 |
| 1 | DS3231 (by Analog Devices, Wilmington, MA, USA) | AliExpress | Real-time Clock module | €1.66 |
| 1 | 0.96-inch OLED Display (by ELEGOO, Shenzhen, China) | AliExpress | Small Display | €2.58 |
| 1 | MicroSD Card Reader (by HiLetgo, Shenzhen, China) | AliExpress | Storage module | €1.78 |
| 1 | DollaTek 18650-Board (by DollaTek, Shenzhen, China) | Amazon | Power management module | €4.72 |
| 3 | 18650 Lithium-Ion Batteries (Price for package of four) | AliExpress | Batteries | €8.77 |
| 1 | HTU21D (by TE Connectivity, Toulouse, France) | AliExpress | Temperature and Humidity sensor | €1.55 |

## 3. Build Instructions

*3.1. P-Tracker Development*

This section outlines the step-by-step process to assemble the P-Tracker device, enabling readers to replicate the hardware. Basic knowledge of electronics and familiarity with soldering, wiring, and microcontroller programming is assumed.

**Step 1: Testing individual components**

First the functionality of the individual components needs to be ensured as described below.

1. Preparation of the Component's ESP32 Development Board: Ensure the microcontroller is functional by running a basic "blink" test [26] using the Arduino IDE.
2. SDS011 PM Sensor: Verify sensor functionality using a test sketch [27] to ensure it outputs data via UART [28].
3. GPS NEO M6 Module: Test the GPS module to ensure it can acquire location data.
4. DS3231 RTC Module: Connect with I2C to verify communication with the ESP32. 0.96-inch OLED Display: Test display output using an I2C display library [29].
5. HTU21D temperature and humidity sensor: Verify the device shows the temperature and humidity [22].

**Step 2: Component layout and connections**

Figure 1 demonstrates the layout and connections between the components of P-tracker.

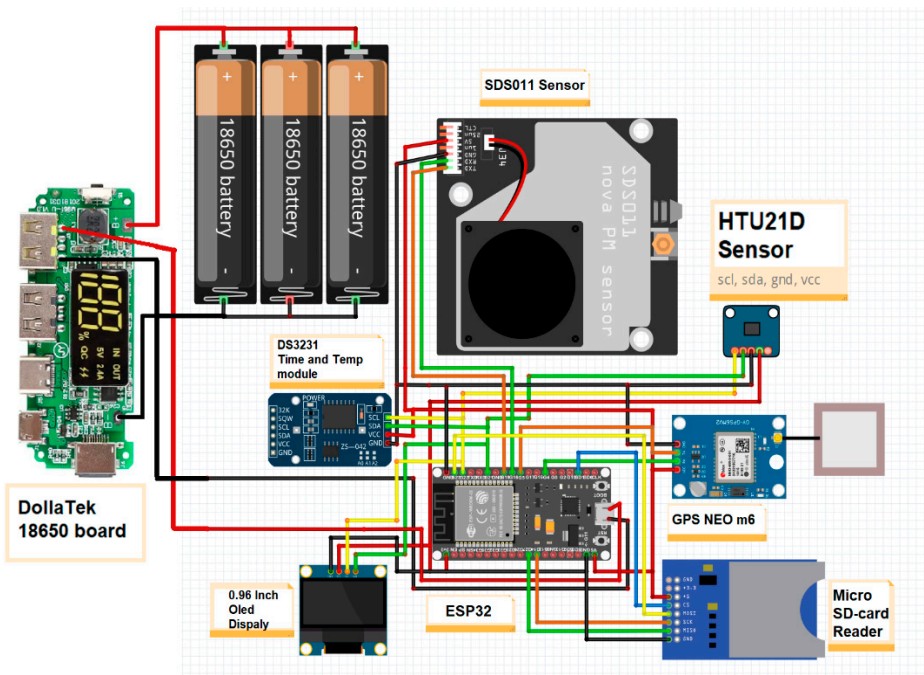

**Figure 1.** Diagram of the P-tracker device.

**Step 3: Connecting components**

1. Assemble the power supply: Connect three 18650 lithium-ion batteries to the DollaTek 18650-Board for power management.
2. The output from the power board (5V and GND) will provide power supply to the ESP32 and other modules.
3. Connect the SDS011 sensor: Wire the SDS011's TX and RX pins to the ESP32's RX and TX pins (GPIO 18 and GPIO 19). Provide 5V from EPS32 and GND to the sensor.
4. Integrate the GPS module: Connect the GPS NEO M6's RX and TX to ESP32 GPIO pins 4 and 5. Provide power 5V from EPS32 and GND.

5.  Connect the DS3231 RTC module: Wire the SDA and SCL pins to ESP32 GPIO pins 21 and 22. Provide 5V power from EPS32 and GND.
6.  Add the OLED display: Connect the OLED's SDA and SCL to the same I2C pins GPIO 21 and 22 as the DS3231. Supply power 3.3V from ESP32 and GND.
7.  Attach the MicroSD card reader: Wire the SPI pins as follows: MOSI to GPIO 23, MISO to GPIO 27, SCK to GPIO 14, CS to GPIO 15, Connect power 5V form EPS32 and GND.
8.  Attach HTU21D Temperature and Humidity module to the 3.3V on ESP32 and GND, and connect the SDA to GPIO 21 and SCL to GPIO 22
9.  Install the ON/OFF button: Wire the button to the DollaTek 18650-Board's power output for enabling/disabling power to the ESP32.
10. Connect HTU21D sensor by connecting to SDA and SCL pins to ESP32 GPIO pins 21 and 22. Provide 5V power and GND.
11. Make sure everything is connected as shown in Table 2. Components list and connectivity.

**Table 2.** Components list and connectivity.

| Components Connected to ESP32 GPIO's | Components Connection | Communication Protocol | Library/Driver |
|---|---|---|---|
| SDS011 (PM2.5/PM10 Sensor) | RX = GPIO 18, TX = GPIO 19 | UART (Serial2) | SdsDustSensor.h |
| GPS NEO M6 v2 | RX = GPIO 4, TX = GPIO 5 | UART (Serial1) | TinyGPS++.h |
| DS3231 RTC (Real-Time Clock) | SDA = GPIO 21, SCL = GPIO 22 | I2C | RTClib.h |
| 0.96-inch OLED Display | SDA = GPIO 21, SCL = GPIO 22 | I2C | Adafruit_SSD1306.h, Wire.h |
| SD Card Reader | CS = GPIO 15, SCK = GPIO 14, MOSI = GPIO 23, MISO = GPIO 27 | SPI | SD.h, SPI.h |
| HTU21D Temperature and Humidity | SDA = GPIO 21, SCL = GPIO 22 | I2C | Adafruit_HTU21DF.h |

**Step 4: Programming ESP32:**

1.  Select the ESP32 in Arduino IDE: Tools—Board—ESP32 Dev Module.
2.  Download the project from Supplementary Materials.
3.  Upload code to Arduino IDE: Open Arduino IDE, File—New, paste code.
4.  Download necessary libraries if required. Upload the Code to ESP32 development module.

**Step 5: Test the Assembly:**

Power up the device and ensure all components initialize correctly. Verify that the OLED displays PM2.5, PM10, Humidity, Temperature, and time. Test Wi-Fi connectivity.

**Step 6: Coding**

Download website's project code.

**Step 7: Server connection**

Set up server connection and database with MongoDB [30] and Heroku [31] (alternatively can set up own server connection on raspberry pi [32–35]).

*3.2. Enclosure*

Taking a sustainable approach, the enclosure was made from a flower spray bottle which functioned as the main outer enclosure for the device, as shown in Figure 2. Guide rails for the inside of the enclosure were made from acrylic leftovers from a local school workshop. The enclosure was made from 100% recycled material. The approach of reuse

and recycling aligns with responsible consumption and production. Sustainable design is crucial in reducing waste, conserving natural resources, and minimizing the environmental impact of manufacturing new products. To minimize the size of enclosure all components were well compacted. The batteries and SDS011 sensor are attached together but separated with 5 mm acrylic. The modules for GPS and the microSD card reader were attached to the top wall of the enclosure. The EPS32 was placed at the bottom of the enclosure to allow space for the wires and the DS3231 module. At the top of the enclosure a place was allocated for the ON/OFF button, 0.96-inch OLED display and input for battery charging. The size of the complete device is 118 mm × 93 mm × 63 mm.

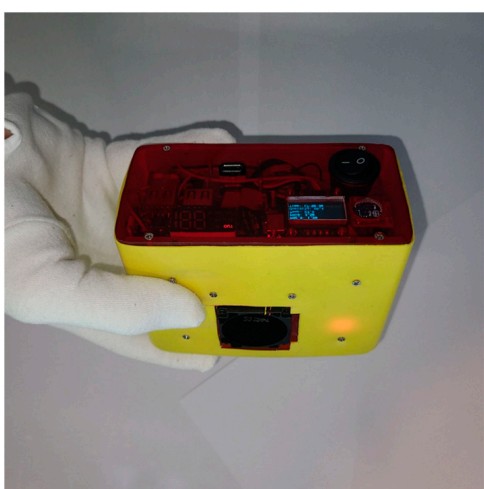

**Figure 2.** P-tracker device in enclosure.

## 4. Operating Instructions

The P-tracker device does not require active user interaction during operation. Once assembled and programmed, it begins measuring, transmitting and saving air quality data automatically. The P-tracker operates by continuously collecting data on particulate matter PM2.5 and PM10, temperature, humidity, time and geographical location. These measurements are captured by the SDS011 sensor and DS3231 module, with location data provided by the GPS module. The core of the device is the ESP32 microcontroller, which manages data acquisition, processing, and transmission. Data are displayed in real time on a compact OLED screen, shown in Figure 3b, and is also stored locally on a microSD card for offline analysis, ensuring data retention even when network connectivity is unavailable. For online data management, the device uses Wi-Fi to transmit the collected data to a remote server. The server, hosted on Heroku [31], a cloud platform that simplifies the deployment and management of applications, runs a Node.js application. This application handles incoming data via HTTP requests and interacts with a MongoDB database, a NoSQL database that stores the data in a flexible, document-oriented format [30]. MongoDB is particularly suited for handling the time-stamped environmental data, allowing efficient storage and retrieval for analysis. The data are then made accessible through a web interface, where users can view real-time readings, track historical data, and analyze trends, as shown in Figure 3a. This integration of hardware, software, and cloud technologies enables a flow of information from the sensor to the user, providing a comprehensive view of air quality in real time.

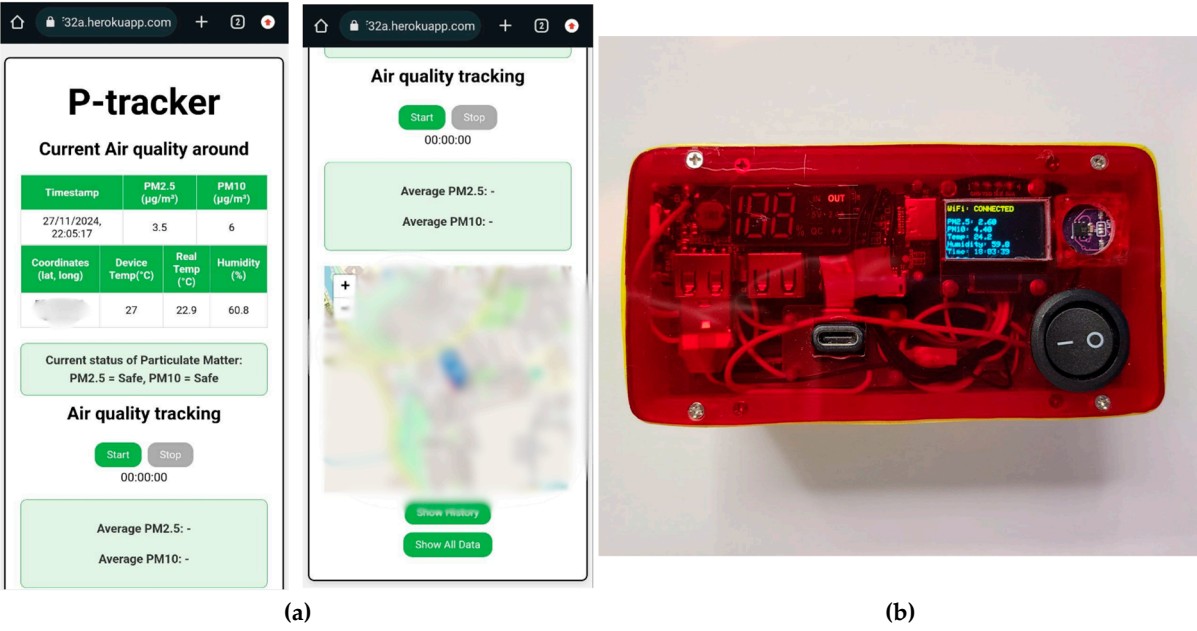

**Figure 3.** P-tracker operating. (**a**) The screenshot of the P-tracker's website open on the mobile phone's browser. (**b**) The assembled P-tracker device with OLED display.

*Potential Safety Hazards*

Ensure the device is not exposed to moisture, water, and direct airflow from fans or a vent. Avoid powering the device from unstable power supplies. Always disconnect power before making hardware changes. Ensure batteries are handled with care. Avoid short-circuiting, overcharging, or exposing batteries to high temperatures. Handle the circuits, components, sensors, and wiring with dry hands to avoid static discharge or accidental damage. Ensure the device is securely mounted or placed to prevent it from falling or getting damaged during use.

## 5. Validation

### 5.1. Commercial Devices

For benchmarking the P-Tracker, several commercial air quality monitoring devices were evaluated based on their capabilities, accessibility, and suitability for comparison. These devices include the AtmoTube Pro (by Atmotech Inc., San Francisco, CA, USA), Flow(by Plume Labs, Paris, France), View Plus (by Airthings, Oslo, Norway), and the Smart Citizen Kit 2.1 (by Fab Lab Barcelona, Barcelona, Spain). Figure 4 shows these devices placed side by side. Additionally, the DustTrak 8530 and Osiris were used as reference measurements for experiments.

The AtmoTube Pro is a compact and portable air quality monitor capable of measuring PM1, PM2.5, PM10, and volatile organic compounds (VOCs). It uses laser-based particle sensing combined with metal-oxide gas sensors [23]. The device also tracks environmental parameters such as temperature, humidity, and atmospheric pressure. Data from the AtmoTube Pro is transmitted to a smartphone app via Bluetooth, providing users with real-time monitoring and customizable pollution alerts [23]. This device is particularly effective for individuals tracking their personal exposure to pollutants across various environments. However, its reliance on a smartphone for data access can be limited, and its sampling interval of one minute may not adequately capture rapid pollution fluctuations.

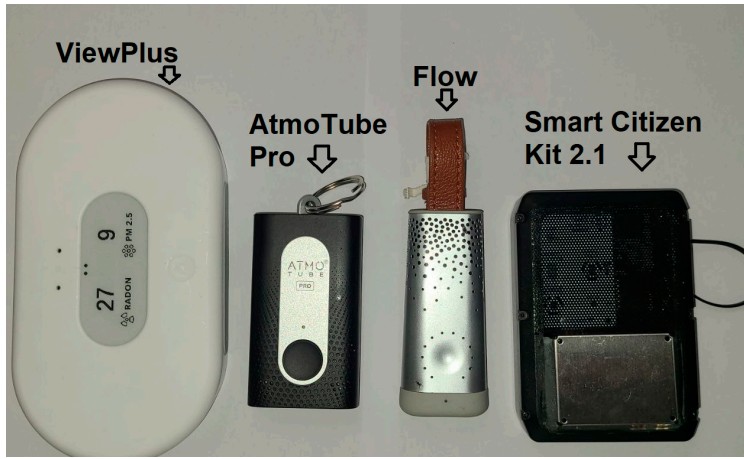

**Figure 4.** Commercial devices.

Smart Citizen Kit 2.1, in contrast, is a stationary, open-source environmental monitoring platform designed for citizen science projects. It measures PM2.5, PM10, VOCs, and $CO_2$, along with additional parameters like temperature, humidity, and noise levels [25]. This device connects to the internet via Wi-Fi, allowing users to access real-time and historical data through the Smart Citizen platform. While the kit is modular and ideal for educational and research purposes, its stationary nature makes it less suitable for applications requiring mobility or personal exposure tracking.

Flow by Plume Labs is a highly portable device designed for urban travel and individuals who prioritize mobility. It measures PM1, PM2.5, PM10, $NO_2$, and VOCs using a combination of laser scattering sensors [24]. The Flow device transmits data to a smartphone app, offering detailed pollution exposure analytics and guidance for daily activities. Despite its small design and robust app interface, it is operating on Bluetooth connectivity, and a one-minute minimum sampling interval may limit its performance in dynamic environments. Since 2023 the Flow device is no longer being sold by Plume Labs which means that unfortunately there is currently no customer support for these devices.

The View Plus by Airthings is an advanced indoor air quality monitor, offering measurements of PM2.5, $CO_2$, VOCs, temperature, humidity, and radon. This device connects to the Airthings app via Wi-Fi, providing comprehensive data visualization and actionable health insights [36]. Its primary focus is on stationary indoor environments, making it a reliable tool for home air quality monitoring but less practical for mobile or outdoor monitoring needs.

DustTrak 8530 serves as the gold standard for evaluating the P-Tracker. This industrial-grade device measures PM1, PM2.5, PM4, PM10, and total particulate mass concentrations with exceptional precision. Utilizing a 90° light scattering laser photometer, it provides real-time data across a wide range of concentrations, from 0.001 to 400 mg/m$^3$ [37]. Its robust design and calibration ensure high accuracy, making it the benchmark for comparative studies. However, its large size, high cost, and stationary nature limit its practicality for everyday use. These devices represent a diverse range of capabilities and approaches to air quality monitoring. The AtmoTube, Flow, and P-Tracker emphasize portability and real-time monitoring, while the View Plus and Smart Citizen Kit focus on stationary and comprehensive environmental assessments. DustTrak provides a reliable reference for validating the accuracy of these devices and the P-Tracker, demonstrating the potential of low-cost alternatives for effective air quality monitoring.

The Osiris air monitoring device is an accurate instrument designed for measuring particulate matter, including PM1, PM2.5, PM10, and total suspended particles (TSP), with a resolution of 0.1 µg/m$^3$ [38]. Certified by the Environment Agency's MCERTS program, the

Osiris guarantees reliable and precise data collection, making it suitable for both short- and long-term air quality studies [38]. Its design allows for use in diverse settings, from urban pollution hotspots to industrial sites, and it can operate as a portable device or in semi-permanent installations. Using a specially developed nephelometer, the Osiris continuously draws air samples through a laser beam, enabling particle analysis. The device also offers flexibility in power sources, supporting battery, solar, or wind power, allowing off-grid deployments. Table 3 shows a comparison of the specifications of these devices.

**Table 3.** Specifications of the devices.

|  | DustTrak | Atmotube | Ptracker | Flow | View Plus | Smart Citizen | Osiris |
|---|---|---|---|---|---|---|---|
| **Sensor model** | not disclosed | SPS30 | SDS011 | Sensor built by Plume Labs | not disclosed | PMS5003 | not disclosed |
| **Sensor technology** | 90° Light scattering | Laser scattering | Laser scattering | Laser scattering | Laser scattering | Laser scattering | Laser beam |
| **Sampling time** | Set by user | 1 min–15 min | Set by user | 1 min | 5 min or 10 min | 5 min | 15 min |
| **Device type** | Portable | Portable | Portable | Portable | Stationary but has batteries | Stationary but has battery | Stationary |
| **Precision** | 0.1% | 10% | 10–15% | not disclosed | 10% | 10% | not disclosed |

*5.2. Humidity Correction Algorithms*

To accurately represent and analyze the collected data from the four tests, a series of data processing steps were implemented. First, a correction algorithm was applied to adjust raw sensor readings for potential inaccuracies due to factors of relative humidity, and sensor placement. Based on studies by Airgradient, the following formulas were applied to the raw data [39]:

**AGraw < 30:**
**PM2.5 = (0.524 AGraw)—(0.0862 RHraw) + 5.75**

**30 $\leq$ AGraw < 50:**
**PM2.5 = (0.786 * (AGraw/20−3/2) + 0.524 * (1 − (AGraw/20−3/2))) * AGraw − (0.0862 * RHraw) + 5.75**

**50 $\leq$ AGraw < 210:**
**PM2.5 = (0.786 * AGraw) − (0.0862 * RHraw) + 5.75**

**210 $\leq$ AGraw < 260:**
**PM2.5 = [0.69 * (AGraw/50−21/5) + 0.786 * (1 − (AGraw/50−21/5))] * AGraw—[0.0862 * RHraw * (1 − (AGraw/50−21/5))] + [2.966 * (AGraw/50−21/5)] + [5.75 * (1 − (AGraw/50−21/5))] + [8.84 * (10$^{-4}$) * AGraw$^2$ * (AGraw/50−21/5)]**

**260 $\leq$ AGraw:**
**PM2.5 = 2.966 + (0.69 * AGraw) + (8.84 * 10$^{-4}$ * AGraw$^2$)**

**AGraw = Sensor readings for PM2.5, RHraw = Relative humidity**

Humidity plays a significant role in the measurement of particulate matter (PM) concentrations, particularly affecting sensors that utilize laser scattering technology. In environments with high relative humidity (RH), airborne particles can absorb moisture, leading to hygroscopic growth, altering their size [40]. This increase in particle size enhances their light-scattering properties, potentially causing laser-based sensors to overestimate PM concentrations. The degree of this effect varies with particle composition and ambient humidity levels. For instance, studies have shown that at elevated RH levels, especially above 85%, the response of these sensors can be significantly influenced, leading to inaccuracies in PM readings [40].

The device Testo 615 (Testo SE & Co. KGaA, Lenzkirch, Germany) was utilized in this study to measure relative humidity, a critical parameter for applying correction algorithms to PM2.5 data. This device provides humidity readings and temperature measurements, which are essential for understanding environmental influences on particulate matter readings. The Testo 615 operates with high accuracy and is ideal for both short-term and long-term monitoring, making it suitable for various environments [41]. Its reliable and consistent performance ensured accurate data collection, enabling the adjustment of PM2.5 measurements for humidity-related effects, such as hygroscopic particle growth, which can impact the accuracy of laser scattering sensors. The Testo 615 was used to ensure the reliability of the air quality data obtained during the tests.

Following the application of the correction algorithm, the data were smoothed using a 15 min moving average filter to compensate for different sampling times from all devices. Given that each test lasted one hour, this averaging process reduced noise and outliers in the data, providing a clearer representation of the overall trends and pollutant levels. This was particularly important for capturing gradual changes in particulate matter concentrations while minimizing the impact of short-term fluctuations.

Finally, a Gaussian filter was applied to the graphs to smooth the lines. The Gaussian filter enhanced the smoothness of the resulting trends by reducing spikes and variations [42].

### 5.3. Results

To evaluate the performance of the P-Tracker against commercial devices and a reference standard, four distinct tests were conducted under controlled and real-world conditions. These tests were designed to assess the accuracy, reliability, and responsiveness of the devices in measuring particulate matter PM2.5 concentrations. Analysis of the experimental results only focused on particulate matter PM2.5 as some devices do not measure PM10, and furthermore PM2.5 is of greater concern given its potential to penetrate into the human body.

### 5.4. Test 1, Paper Combustion

The first test involved a controlled environment where all devices, including the P-Tracker, AtmoTube Pro, Flow by Plume Labs, Smart Citizen Kit 2.1, and DustTrak 8530, were placed inside a sealed 50 L plastic bag. Within the bag, a small piece of paper measuring 20 mm by 20 mm was burned to create a high concentration of combustion aerosols. The test ran for one hour, allowing the sensors to respond to the rising and stabilizing levels of particulate matter. Table 4 below shows the PM2.5 readings from all devices during the test, averaged over 15-minute intervals. The DustTrak 8530 was used as the gold standard reference due to its industrial-grade precision. This test was crucial in determining the devices' ability to detect sudden changes in PM2.5 levels in a highly polluted microenvironment.

**Table 4.** Particulate matter PM2.5 readings moving average 15 min.

| | DustTrak | Atmotube | Ptracker | Flow | View Plus | Smart Citizen |
|---|---|---|---|---|---|---|
| Value at 15th min, PM 2.5 µg/m$^3$ | 1,142,773.63 | 1398.90 | 1192.07 | 32.33 | 338.29 | 3260.16 |
| Value at 30th min PM 2.5 µg/m$^3$ | 2899.07 | 406.12 | 385.63 | 29.36 | 6.05 | 796.96 |
| Value at 45th min PM 2.5 µg/m$^3$ | 31.09 | 8.01 | 13.08 | 2.06 | 2.78 | 48.14 |
| Value at 60th min PM 2.5 µg/m$^3$ | 2.69 | 1.93 | 1.68 | 2.37 | 3.01 | 1.62 |

The correlation coefficients between each device and DustTrak were calculated for three specific time intervals: 15th to 60th minute, 30th to 60th minute, and 45th to 60th minute, as shown in Table 5. During the first 15 min the concentrations were beyond the range of sensors and vastly outside of the range of expected use for a personal sensing device. The AtmoTube Pro demonstrated a strong Pearson correlation across all time ranges, starting at 0.76 for the 15th to 60th minute and achieving perfect correlation 1.00 in the final interval 45th to 60th minute within lower ranges of PM 2.5. This highlights its ability to adapt and align with the reference standard over time. Similarly, the P-Tracker displayed consistently high correlations, starting at 0.73 for the first range and increasing to 0.99 for the final two intervals, showcasing its reliability and effectiveness in tracking PM2.5 levels.

**Table 5.** Test 1, correlation results.

| Devices | Relative Humidity | Correlation at Range 15$^{th}$ min−60$^{th}$ min | Correlation at Range 30$^{th}$ min−60$^{th}$ min | Correlation at Range 45$^{th}$ min−60$^{th}$ min |
|---|---|---|---|---|
| AtmoTube vs. DustTrak | 50–60% | 0.76 | 0.99 | 1 |
| P-tracker vs. DustTrak | 50–60% | 0.73 | 0.97 | 0.99 |
| Flow vs. DustTrak | 50–60% | 0.44 | 0.94 | 0 |
| View Plus vs. DustTrak | 50–60% | 0.90 | 0.85 | 0 |
| Smart Citizen vs. DustTrak | 50–60% | 0.70 | 0.85 | 0.88 |

The Flow by Plume Labs showed a weaker initial correlation of 0.44 during the first range but improved to 0.94 in the 30th to 60th minute interval. However, it failed to register any meaningful correlation in the final interval. The View Plus by Airthings initially achieved the highest correlation 0.90 during the 15th to 60th minute interval but declined to 0.85 and 0.00 in the subsequent intervals, suggesting variability in its performance. The Smart Citizen Kit showed moderate correlation values, starting at 0.70 in the first interval and gradually increasing to 0.88 in the final interval, demonstrating acceptable but less robust performance compared to the AtmoTube and P-Tracker.

Overall, the correlation results from the paper combustion test demonstrate the reliability of the AtmoTube Pro and P-Tracker in closely aligning with the DustTrak 8530, making them effective options for monitoring PM2.5 concentrations in controlled high-pollution environments. The variability observed in other devices, such as Flow and View Plus, highlights the importance of calibration and response stability when assessing particulate matter levels.

In Table 6, the statistical results for the combustion Test 1 are presented, focusing on the mean, median, mode, and standard deviation for PM2.5 measurements recorded from the 30th minute of the test. The DustTrak, used as the reference device, displayed the highest mean value of 315.60 μg/m$^3$ with a standard deviation of 656.69 μg/m$^3$, reflecting the dynamic nature of particulate matter during combustion.

**Table 6.** Test 1, mean, median, mode and standard deviation results.

| Devices | Mean | Median | Mode | Standard Deviation |
|---|---|---|---|---|
| DustTrak | 315.60 | 31.09 | 2.69 | 656.69 |
| AtmoTube | 53.89 | 8.01 | 1.93 | 97.13 |
| P-tracker | 60.81 | 13.08 | 1.68 | 96.72 |
| Flow | 6.63 | 2.32 | 1.70 | 7.88 |
| View Plus | 3.13 | 2.75 | 3.01 | 1.19 |
| Smart Citizen | 210.71 | 48.14 | 4.24 | 282.26 |

Among the tested devices, the P-Tracker exhibited a mean of 60.81 μg/m$^3$ and a standard deviation of 96.72 μg/m$^3$, aligning closely with the AtmoTube Pro, which recorded a mean of 53.89 μg/m$^3$. and standard deviation of 97.13 μg/m$^3$. The Smart Citizen Kit showed a higher mean of 210.71 μg/m$^3$, but its significantly large standard deviation of 282.26 μg/m$^3$ indicates less consistent readings. The Flow and View Plus devices reported substantially lower mean values 6.63 μg/m$^3$ and 3.13 μg/m$^3$, suggesting potential underestimation of particulate concentrations in this controlled high-pollution environment.

The graph in Figure 5 illustrates the PM2.5 measurements during paper combustion Test 1, comparing six devices, including the DustTrak 8530 P-Tracker, AtmoTube Pro, Flow, View Plus, and Smart Citizen Kit. The *y*-axis shows PM2.5 concentrations 0–2000 μg/m$^3$, while the *x*-axis represents time over a one-hour period.

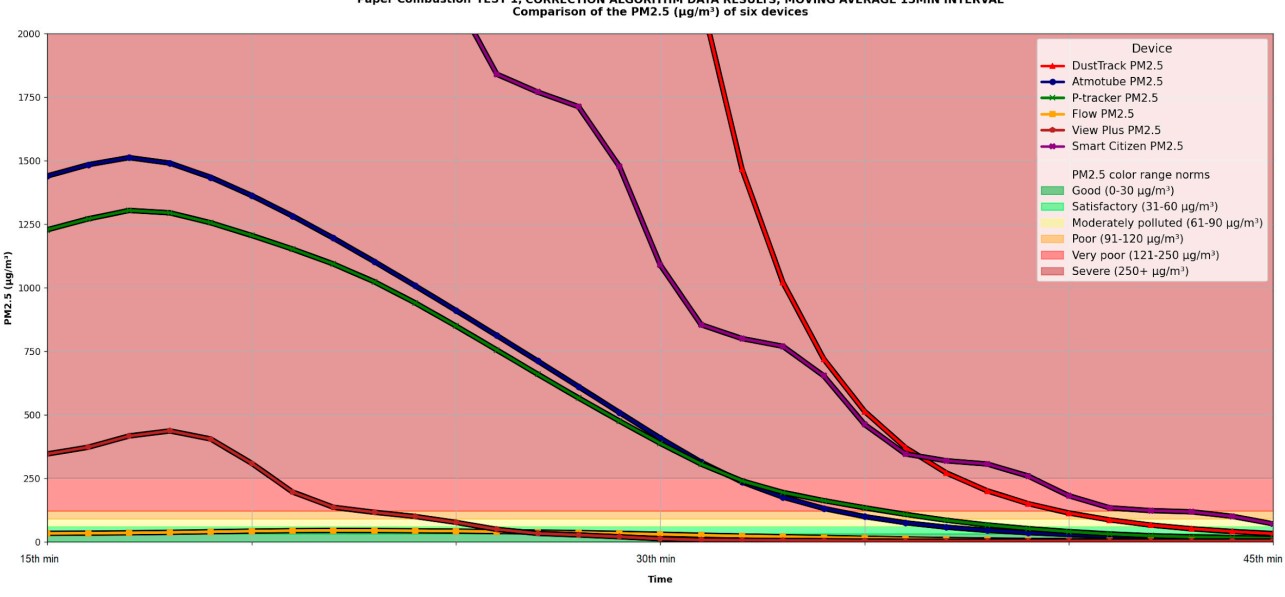

**Figure 5.** Paper combustion test—15th to 45th minute timeline.

The DustTrak records the highest PM2.5 levels, peaking more than 2000 μg/m$^3$, with the P-Tracker and AtmoTube Pro closely following its trend, demonstrating strong agreement. The Smart Citizen Kit aligns moderately well but exhibits a slower response

during peak concentrations. In contrast, the Flow and View Plus significantly underestimate PM2.5 levels, particularly during the combustion phase, rarely exceeding the "Very Poor" range 121–250 µg/m$^3$.

Color-coded air quality norms [43] in the background provide context for pollution severity. All devices capture the eventual decline in PM2.5 levels, with the P-Tracker and AtmoTube Pro standing out for their consistent performance relative to the DustTrak device.

Figure 6 provides a closer look at the last 15 min of the paper combustion test, focusing on PM2.5 concentrations capped at 20 µg/m$^3$. The DustTrak 8530, serving as the reference, consistently records higher PM2.5 levels throughout the test, demonstrating its precise response to particulate matter generated during combustion. The P-Tracker closely follows DustTrak's trend, showcasing strong reliability and sensitivity, with minimal deviations from the reference measurements. The AtmoTube Pro also aligns well with DustTrak, maintaining consistent readings that reflect the general decay trend of PM2.5 concentrations over time. This highlights its capability for accurate tracking of particulate matter in controlled environments. The Smart Citizen Kit follows a similar pattern but exhibits delayed responses to changes in PM2.5 levels, indicating slower adaptability compared to the AtmoTube Pro and P-Tracker. In contrast, the Flow and View Plus devices continue to significantly underestimate PM2.5 concentrations, rarely exceeding the lower ranges of particulate matter levels. Their consistent underreporting highlights limitations in their ability to detect small but important variations during high-pollution events. Overall, Figure 5. shows the strong performance and reliability of the P-Tracker and AtmoTube Pro in capturing particulate matter during combustion events. Meanwhile, the limited accuracy of the Flow and View Plus devices highlights the challenges of ensuring precision and sensitivity in stationary and portable air quality monitoring devices, particularly in dynamic, high-pollution environments.

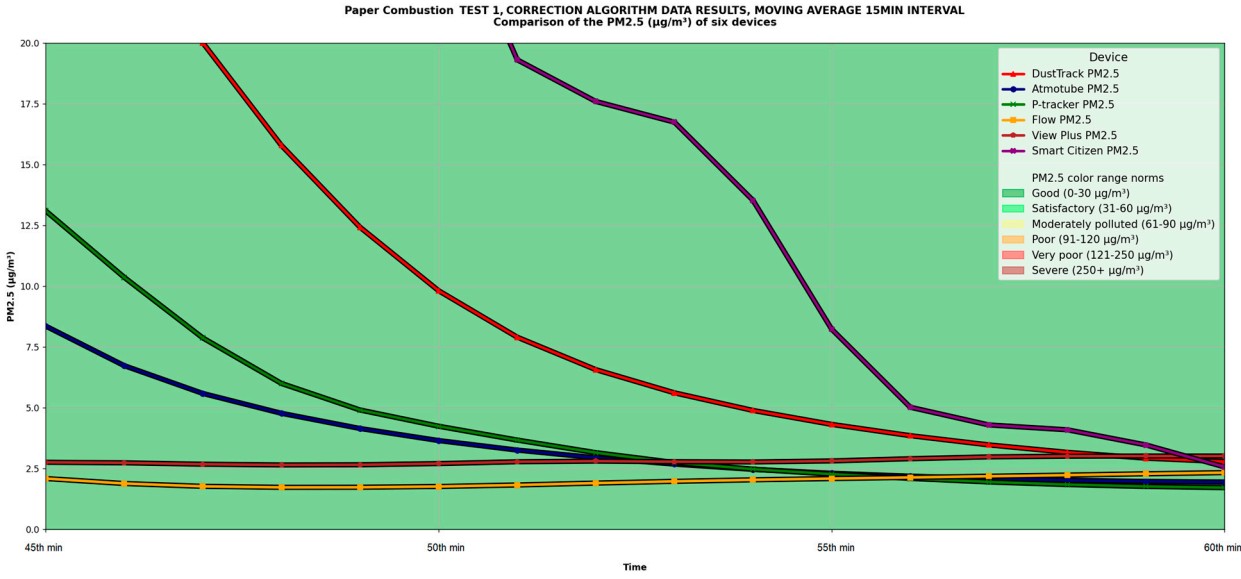

**Figure 6.** Paper combustion test, final 15 min of test period.

### 5.5. Test 2, Comparing Devices with Osiris

The second test was conducted outdoors in a real-world setting, with all devices placed 10–20 m to Osiris air quality monitoring station which located in a local library in Drumcondra, Dublin. The Osiris is attached to the library's external [44] wall approximately five meters from ground. As an official air quality monitoring station, this served as the reference device. Over the course of an hour, the sensors were exposed to ambient air conditions, including varying levels of vehicular emissions, dust, and other pollutants.

This test aimed to show how portable and stationary devices, including the P-Tracker, could match stationary reference devices in capturing fluctuating outdoor pollution levels. The data highlighted the challenges of dynamic outdoor environments, such as wind and uneven pollutant distribution and humidity.

The graph in Figure 7 illustrates PM2.5 measurements recorded during Test 2, where portable devices were compared to the stationary Osiris air quality monitoring station under outdoor conditions at the Drumcondra library. The data reflects real-world ambient air conditions, including fluctuating vehicular emissions and environmental factors like wind and humidity. The AtmoTube Pro shows consistent performance, maintaining a slightly higher PM2.5 concentration compared to the Osiris. This alignment suggests its reliability in outdoor scenarios. Similarly, the P-Tracker closely follows the trend of the Osiris. Flow by Plume Labs also exhibits a pattern similar to the AtmoTube, but its readings demonstrate minor fluctuations, highlighting moderate reliability in dynamic outdoor conditions. In contrast, View Plus displays significant variability, with its PM2.5 values deviating sharply at certain points, particularly around the midpoint of the test. This irregular performance suggests potential sensitivity to environmental factors or limitations in calibration. The Smart Citizen Kit, however, fails to capture meaningful data, with its readings consistently remaining at zero throughout the test, highlighting its unsuitability for outdoor air quality monitoring in this context. Overall, the graph demonstrates the AtmoTube Pro and P-Tracker as the most consistent performers, closely aligning with the Osiris reference, while Flow provides moderate reliability.

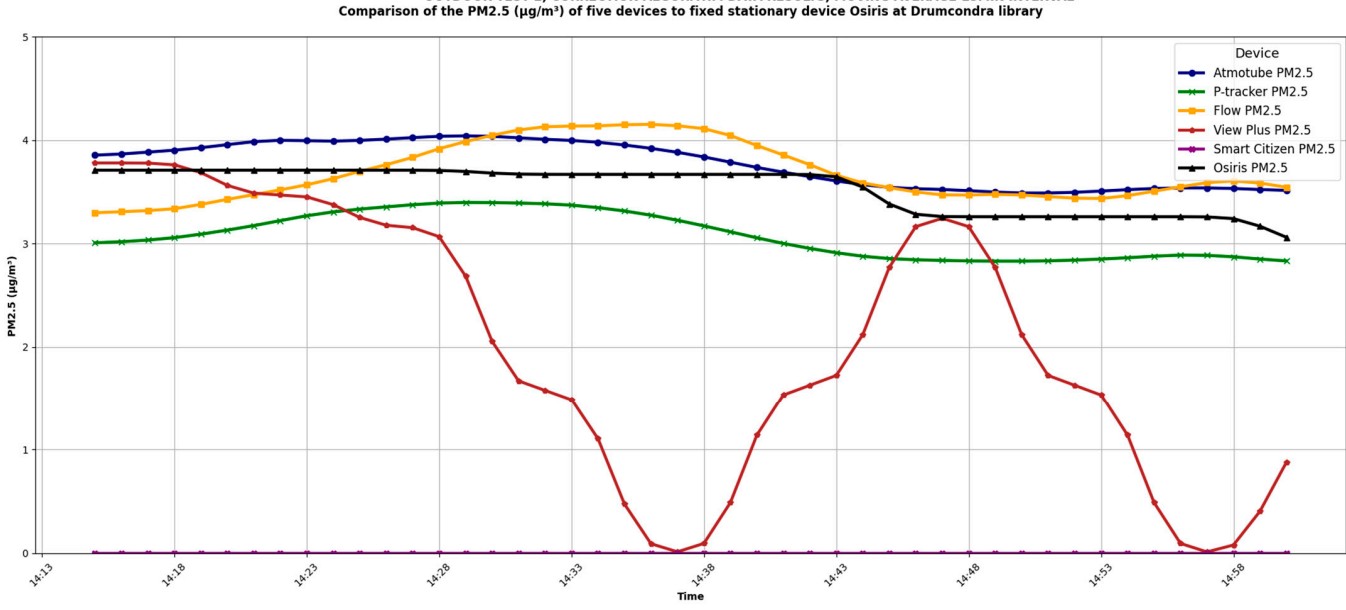

**Figure 7.** Test 2, Comparison the mobile devices with Osiris.

The results of Test 2, as summarized in Table 7, provide insights into the performance of the portable devices compared to the Osiris fixed station. The AtmoTube Pro recorded the highest mean value of 3.77 µg/m$^3$, slightly exceeding the reference device's mean of 3.53 µg/m$^3$. It also exhibited a low standard deviation of 0.22, indicating consistent readings that align closely with the Osiris station. The P-Tracker displayed a mean value of 3.08 µg/m$^3$, with a standard deviation of 0.22, demonstrating both reliability and moderate agreement with the Osiris. Its median value of 3.04 µg/m$^3$ further supports its consistent performance in capturing outdoor PM2.5 levels under fluctuating conditions. Flow recorded a slightly higher mean value of 3.69 µg/m$^3$, close to the AtmoTube, but with a higher standard deviation of 0.28. This suggests less stability in its readings compared to

the AtmoTube and P-Tracker. View Plus recorded the lowest mean value of 2.04 $\mu g/m^3$, with a significant standard deviation of 1.37, indicating considerable variability in its measurements and limited agreement with the reference device. The Smart Citizen Kit failed to record any meaningful data during the test, showing mean, median, and mode values of 0.00 $\mu g/m^3$. This highlights potential limitations in its sensitivity or configuration for outdoor monitoring scenarios. Overall, the results indicate that the AtmoTube Pro and P-Tracker provided the most reliable and consistent performance in outdoor conditions, aligning closely with the Osiris fixed station. In contrast, Flow and View Plus displayed greater variability, compared to the Smart Citizen Kit.

**Table 7.** Test 2, mean, median, mode and standard deviation results.

| Device | Mean | Median | Mode | Standard Deviation |
|--------|------|--------|------|--------------------|
| Osiris | 3.53 | 3.67 | 3.26 | 0.22 |
| AtmoTube | 3.77 | 3.84 | 3.54 | 0.22 |
| P-tracker | 3.08 | 3.04 | 2.83 | 0.22 |
| Flow | 3.69 | 3.58 | 3.45 | 0.28 |
| View Plus | 2.04 | 1.63 | 0.00 | 1.37 |

Figure 8 shows the correlation between each device against the Osiris reference device. The data points of PM2.5 are shown as colored dots, where red dots are for Osiris, blue for AtmoTube, green for P-tracker, yellow for Flow and light purple for View Plus. The red line in each plot represents linear regression, it shows the best-fit line through the data points for each device compared to the reference device Osiris. The data in Table 8 for Test 2 presents the relationship between the portable devices and the Osiris fixed station under outdoor conditions, with relative humidity ranging between 30 and 41%. The AtmoTube Pro shows the strongest performance, achieving a Pearson correlation [45] of 0.86 and a Spearman correlation [46] of 0.81, reflecting its high accuracy and reliability in capturing PM2.5 concentrations. With a slope [47] of 0.84 and a low root mean square error (RMSE [48]) of 0.26, the AtmoTube Pro closely aligns with the Osiris, demonstrating its capability for dynamic outdoor monitoring. The P-Tracker follows with a Pearson correlation of 0.77 and a Spearman correlation of 0.73, showing consistent and reliable agreement with the Osiris. Its slope of 0.75 and RMSE of 0.48 highlight its strong performance as a cost-effective alternative, but it is slightly less precise than the AtmoTube Pro. Flow by Plume Labs exhibits a moderate Pearson correlation of 0.41 but a weaker Spearman correlation of 0.07, indicating limited consistency in ranked pollutant measurements. Its slope of 0.53 and RMSE of 0.32 suggest some variability in aligning with the Osiris under fluctuating conditions. View Plus shows the lowest Pearson correlation of 0.28 but a higher Spearman correlation of 0.54, indicating that while it struggles with absolute measurements, it captures some trends in ranked data. Its slope of 1.75 and a high RMSE of 1.98 indicate significant deviations, particularly in outdoor environments. The Smart Citizen Kit recorded zero values for all metrics, including Pearson and Spearman correlations, slope, and RMSE. In summary, the AtmoTube Pro and P-Tracker show the most reliable and accurate performance in outdoor environments, with high correlations and low errors. Flow and View Plus show limited reliability with greater variability.

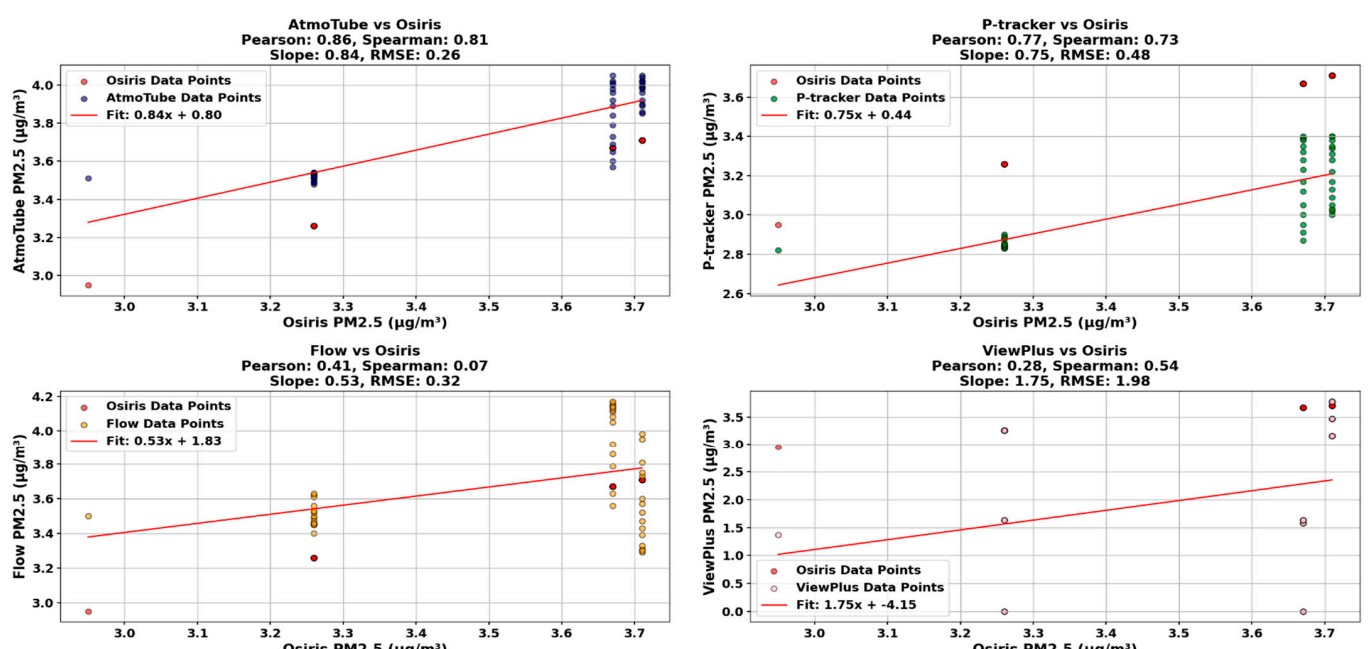

**Figure 8.** Comparative analysis of PM2.5 measurements across different devices against Osiris reference.

**Table 8.** Comparative analysis of mobile devices and Osiris.

| Devices | Relative Humidity | Pearson Correlation | Spearman Correlation | Slope | Root Mean Square Error |
|---|---|---|---|---|---|
| AtmoTube vs. Osiris | 30–41% | 0.86 | 0.81 | 0.84 | 0.26 |
| P-tracker vs. Osiris | 30–41% | 0.77 | 0.73 | 0.75 | 0.48 |
| Flow vs. Osiris | 30–41% | 0.41 | 0.07 | 0.53 | 0.32 |
| View Plus vs. Osiris | 30–41% | 0.28 | 0.54 | 1.75 | 1.98 |
| Smart Citizen vs. Osiris | 30–41% | 0.00 | 0.00 | 0.00 | 0.00 |

*5.6. Test 3, Indoor Air Quality*

The third test focused on indoor air quality monitoring. All devices were placed in a controlled indoor environment, where they recorded PM2.5 levels for one hour. This test was designed to evaluate the devices' sensitivity to typical indoor pollutants, such as dust, and particles from HVAC systems. The controlled nature of this environment allowed for a more stable comparison of sensor accuracy and consistency. Results showed that all devices provided relatively the same trend but still had differences in particulate matter readings.

The results of Test 3, presented in Table 9, provide detailed insights into the performance of each device in a controlled indoor environment. The AtmoTube Pro recorded the highest mean PM2.5 value of 2.34 μg/m$^3$ with a median of 2.37 μg/m$^3$ and a standard deviation of 0.11, indicating consistent and stable readings. This highlights its reliability for indoor air quality monitoring. The P-Tracker closely followed with a mean and median of 1.59 μg/m$^3$ and a similarly low standard deviation of 0.11, demonstrating both accuracy and consistency in its measurements. Flow by Plume Labs showed a slightly higher mean of 1.74 μg/m$^3$ with a median of 1.69 μg/m$^3$, but its standard deviation of 0.23 reflects greater variability compared to the AtmoTube and P-Tracker. The View Plus device recorded a mean of 2.17 μg/m$^3$ and the highest standard deviation among the devices at 0.87. This indicates significant fluctuations in its readings, which could limit its reliability in detecting stable indoor pollutant levels. Meanwhile, the Smart Citizen Kit failed to record any PM2.5 readings, showing consistent zero values for all metrics, which highlights its limitations in

detecting particulate matter in low levels of PM2.5, highlighting the challenges of having a versatile device with sufficient sensitivity. Overall, the results indicate that the AtmoTube Pro and P-Tracker delivered the most reliable and consistent performance during indoor air quality monitoring, while the Flow and View Plus showed moderate reliability with greater variability.

**Table 9.** Test 3, mean, median, mode and standard deviation results.

| Device | Mean | Median | Mode | Standard Deviation |
|---|---|---|---|---|
| AtmoTube | 2.34 | 2.37 | 2.47 | 0.11 |
| P-tracker | 1.59 | 1.59 | 1.59 | 0.11 |
| Flow | 1.74 | 1.69 | 1.46 | 0.23 |
| View Plus | 2.17 | 2.52 | 0.00 | 0.87 |

The graph in Figure 9 for Indoor Test 3 presents the PM2.5 measurements captured by the devices over the course of one hour in a controlled indoor environment. The controlled setting allowed for stable conditions to evaluate the sensitivity and consistency of each device. The AtmoTube Pro displayed stable and consistent readings throughout the test, maintaining PM2.5 levels slightly above 2 µg/m$^3$. Its steady trend aligns with expectations for a reliable device in controlled environments. Similarly, the P-Tracker recorded slightly lower PM2.5 levels, consistently averaging near 1.5 µg/m$^3$, showcasing its reliable performance in indoor conditions with minimal variability. The Flow device also performed steadily, with readings slightly above the P-Tracker, hovering near 1.7 µg/m$^3$. While its measurements were consistent, they were slightly elevated compared to the P-Tracker, which may indicate a minor bias in detecting particulate matter. The View Plus exhibited more variability, with readings starting near 2.5 µg/m$^3$ but showing a sharp decline toward the end of the test. The Smart Citizen Kit failed to register any meaningful PM2.5 data, maintaining a flat line at 0 µg/m$^3$ throughout the test, reflecting its inability to detect indoor particulate matter effectively. This highlights significant limitations in its application for indoor air quality monitoring. Overall, the AtmoTube Pro and P-Tracker demonstrated the most consistent and reliable performance in measuring PM2.5 levels.

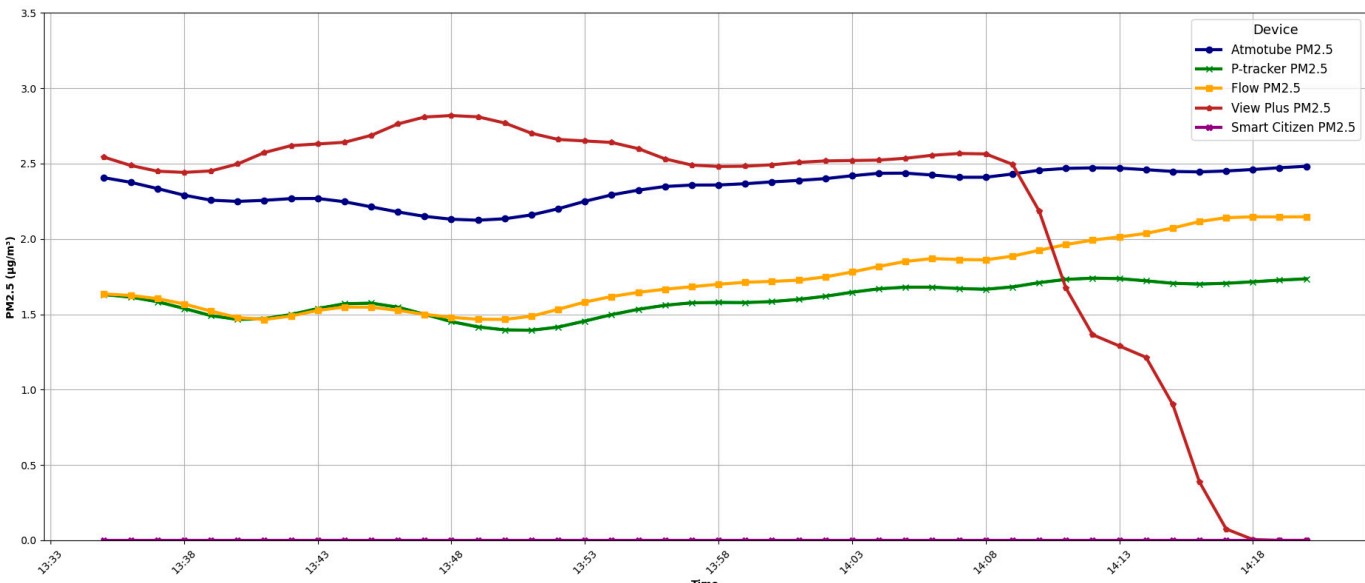

**Figure 9.** Test 3, indoor air quality test.

*5.7. Test 4, Real Environment Air Quality*

The fourth and final test simulated a real-world kitchen environment, where indoor air quality is often affected by cooking activities. Over the span of one hour, sensors recorded particulate matter concentrations as cooking fumes and other aerosols were generated. This test replicated the everyday scenarios where portable air quality monitors are most likely to be of benefit. The P-Tracker and other devices demonstrated its practicality by effectively capturing the spikes in PM2.5 levels during active cooking. In contrast, some commercial devices, such as the Flow and Smart Citizen Kit, exhibited slower response times or lower sensitivity to rapid changes in pollutant levels.

The results from Test 4 in Table 10, which focused on real-environment air quality monitoring in a kitchen setting. The AtmoTube Pro recorded the highest reliability, with a mean PM2.5 concentration of 2.88 µg/m$^3$ and a standard deviation of 0.14, indicating highly consistent readings during the test. Its median and mode values, both at 2.91 µg/m$^3$, reflect stable and accurate performance in tracking kitchen-related air quality changes. The P-Tracker performed well, with a mean value of 1.64 µg/m$^3$ and a standard deviation of 0.19, showcasing consistent and reliable measurements. Its alignment with median and mode values highlights its ability to track PM2.5 levels effectively in dynamic conditions. Flow by Plume Labs showed a slightly lower mean of 1.59 µg/m$^3$ and a higher standard deviation of 0.39, indicating greater variability in its measurements compared to the AtmoTube and P-Tracker. The View Plus recorded the highest mean PM2.5 value among all devices at 4.11 µg/m$^3$, with a standard deviation of 0.90. This significant variability suggests potential sensitivity to rapid changes or challenges in maintaining stability during fluctuating pollutant levels. The Smart Citizen Kit, with a mean of 0.56 µg/m$^3$ and a standard deviation of 0.30, demonstrated limited sensitivity, underestimating PM2.5 concentrations in this dynamic environment. Overall, the AtmoTube Pro and P-Tracker provided the most consistent and accurate results in the kitchen environment, while the Flow and View Plus exhibited greater variability.

**Table 10.** Test 3, kitchen environment air data collection.

| Device | Mean | Median | Mode | Standard Deviation |
|--------|------|--------|------|---------------------|
| AtmoTube | 2.88 | 2.91 | 2.91 | 0.14 |
| P-tracker | 1.64 | 1.63 | 1.64 | 0.19 |
| Flow | 1.59 | 1.43 | 1.32 | 0.39 |
| View Plus | 4.11 | 4.00 | 2.41 | 0.90 |
| Smart Citizen | 0.56 | 0.59 | 0.87 | 0.30 |

The graph in Figure 10 for Test 4 illustrates the PM2.5 measurements recorded over one hour in a real-world kitchen environment. The test reflects the device's performance in capturing particulate matter generated by typical kitchen activities, such as cooking. The AtmoTube Pro maintained consistent readings, averaging around 3 µg/m$^3$, with minimal fluctuations over time. Its stable trend highlights its reliability in environments with moderate pollutant levels. The P-Tracker displayed slightly lower PM2.5 concentrations, averaging about 1.6 µg/m$^3$. The P-Tracker's consistent and smooth trendline indicates a steady response to kitchen activities with minimal variability. Flow by Plume Labs recorded values close to the P-Tracker, averaging slightly above 1.5 µg/m$^3$ but with minor fluctuations throughout the test. The View Plus, however, reported the highest PM2.5 concentrations, averaging over 4 µg/m$^3$, with noticeable variability. This variability suggests that the View Plus may be overly sensitive to environmental changes, leading to less

stable readings in dynamic conditions. The Smart Citizen Kit recorded significantly lower values, consistently near 0 $\mu g/m^3$, failing to detect the PM2.5 levels effectively. Overall, the AtmoTube Pro and P-Tracker demonstrated the most consistent and reliable performance, closely aligning with expected trends for kitchen-generated particulate matter. Flow provided moderate reliability with slight variability, while the View Plus exhibited excessive sensitivity, and the Smart Citizen Kit proved ineffective in capturing kitchen-related PM2.5 levels.

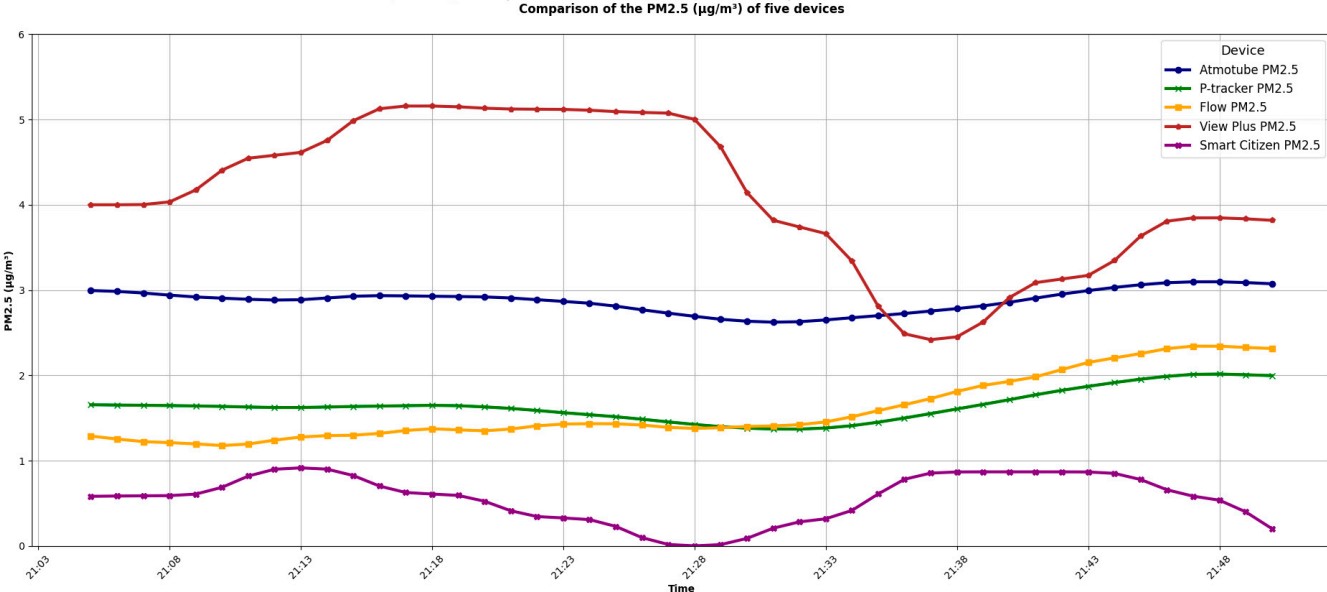

**Figure 10.** Test 4, kitchen air quality test.

## 6. Conclusions

This study evaluated the performance of the custom-built P-Tracker air quality monitoring device against four commercial devices—AtmoTube Pro, Flow by Plume Labs, ViewPlus by Airthings, and Smart Citizen Kit 2.1—across four distinct test environments: a controlled paper combustion test, outdoor monitoring, indoor monitoring, and real-world kitchen conditions. The DustTrak 8530 and the Osiris stationary monitor served as reference standards, providing reliable baselines for comparison. This paper presents the complete design and build instructions to make this device accessible to researchers and communities, thereby enabling hyper-local monitoring of air quality to address the environmental and health impact of air pollution locally and globally. The P-tracker links to a website which allows the communication and visualization of local air quality measurements.

The P-Tracker was evaluated with commercial reference systems in a number of test environments and the device demonstrated consistent performance throughout the tests, showing strong agreement with reference devices in both controlled and dynamic environments. In the paper combustion test, the P-Tracker captured the trend of rising and stabilizing PM2.5 concentrations, achieving high correlations with the DustTrak (up to 0.99). Its ability to accurately track particulate matter during paper combustion test highlights its effectiveness in detecting high-pollution scenarios. During outdoor monitoring, the P-Tracker maintained a solid correlation of 0.77 with the Osiris, showcasing its ability to adapt to fluctuating outdoor conditions, such as wind, vehicular emissions and humidity. In indoor settings, the P-Tracker delivered stable readings with low variability, aligning closely with the AtmoTube Pro and reference device. Its performance in the kitchen test further validated its reliability, as it consistently captured particulate matter fluctuations from cooking activities with minimal deviations.

The AtmoTube Pro consistently emerged as the most reliable commercial device, with high correlations to reference devices in all tests. Its stable performance in both controlled and real-world environments underscore its suitability for personal air quality monitoring. It performed well in capturing dynamic trends and provided accurate and consistent readings across all scenarios. The AtmoTube Pro's close alignment with the P-Tracker in multiple tests validates the potential of the P-Tracker as a cost-effective alternative for air quality monitoring.

Flow by Plume Labs performed moderately well, particularly in outdoor and indoor tests, but exhibited greater variability compared to the AtmoTube Pro and P-Tracker. Its performance was less consistent in controlled and high-pollution environments, such as the paper combustion and kitchen tests, where it underestimated particulate concentrations. The View Plus showed significant variability across all tests, with inconsistent readings and sensitivity issues, particularly in dynamic and high-pollution scenarios. Although its design may focus on other air quality metrics, its limitations in capturing PM2.5 concentrations highlight the need for improved calibration and stability. The Smart Citizen Kit demonstrated severe limitations, failing to provide meaningful data in most scenarios. Its consistent zero readings in outdoor and indoor tests reveal challenges in detecting PM2.5 effectively. While the kit's design may prioritize other environmental parameters, its inability to detect particulate matter renders it unsuitable for applications requiring accurate PM2.5 monitoring.

The P-Tracker is a simple, affordable, and versatile air quality monitoring device that stands out because of its features. One of the abilities is to track location by attaching GPS coordinates to every air quality reading. These readings, along with their precise locations, can be viewed on the custom website. The website displays data and an interactive map, allowing users to see where air quality measurements were taken. By making the data available through an open access website, it ensures that users can easily see and analyze their readings without needing expensive software or technical expertise. The website provides a clear, real-time view of pollution levels, combined with the ability to track these levels geographically. This combination of location-based monitoring and simple data access makes the P-Tracker an ideal choice for personal air quality tracking, environmental research, and projects.

Unlike some commercial devices like the View Plus, which are designed to stay in one place and monitor indoor air quality, the P-Tracker is portable. Its design makes it easy to carry, allowing it to be used in various locations, both indoors and outdoors. This portability allows the P-Tracker to collect data in dynamic environments, such as busy streets, parks, and workplaces, or even while commuting. Stationary devices are limited to providing air quality readings for a single spot, which can miss important variations in polluted areas. The P-Tracker's ability to move with the user and map air quality indoors and outdoors across different environments gives it an advantage for those who want a better understanding of their exposure to pollution as they change location during the day.

Overall, the P-Tracker proved to be a reliable and cost-effective air quality monitoring device, consistently performing close with the AtmoTube Pro. Its robust performance in controlled and real-world scenarios validates its potential as an alternative to commercial devices, particularly for personal and citizen science applications. The study also highlights the variability in commercial device performance, with the AtmoTube Pro standing out as the most consistent and reliable. These findings emphasize the importance of accuracy, stability, and calibration in air quality monitoring devices, particularly for capturing PM2.5 levels in diverse environments.

*Modularity and Future Work*

The current design of the P-tracker provides a fundamental platform for portable air quality devices for measuring PM2.5 and PM10. However, the device was built to allow for future upgrades, since it uses a full size ESP32 microcontroller board with multiple I/O pins and supports multiple communication protocols such as I2C, UART and SPI, this allows integration of additional sensors in the future. This means users can modify and expand the device to measure other air quality parameters, such as volatile organic compounds (VOCs), carbon dioxide ($CO_2$), carbon monoxide (CO), nitrogen dioxide ($NO_2$) and methane (CH4) by connecting MQ-series sensors or ENS160+AHT21 sensors.

The P-tracker is designed in a flexible way that makes it easy to upgrade or replace parts and components without rebuilding the whole device. For example, the SDS011, the main sensor of the device can be replaced with SPS30, a reliable and accurate sensor, which is used in commercial device AtmoTube PRO. The projects code can also be adjusted to support new sensors that may emerge with minimal changes to the existing platform.

Another future enchantment could be the integration into a global map such as the "Sensor.Community" open-source platform for environmental sensor data [49]. This is a website that collects and shares open air quality data worldwide from over 70 countries, to inform communities and involve citizens in the design of their cities. This article aims to contribute to the maker community in providing open-source designs to support engaged research for the benefit of communities and wider society.

**Supplementary Materials:** The following supporting information can be downloaded at: https://www.mdpi.com/article/10.3390/hardware3040012/s1.

| Name | Type | Description |
|---|---|---|
| ESP32_code | Folder | Folder containing code for ESP32 |
| ESP32_code_P_tracker.ino | Arduino ino | Arduino ino file for ESP32 |
| Components Diagram.png | PNG | Connected components diagram |
| Website_SET_UP_instructions.pdf | PDF | Instructions PDF file for the website set up |
| WEBSITE_code | Folder | Folder containing code for Website |
| js | Folder | Folder inside WEBSITE_code directory for java scripts |
| stylesheets | Folder | Folder inside WEBSITE_code directory for CSS files |
| index.html | HTML | Main HTML file in the WEBSITE_code directory for the website front end. |
| server.js | JavaScript | Backend JavaScript file in the WEBSITE_code directory that starts the web server, handles API requests, stores data in MongoDB, and manages real-time updates. |
| readme.txt | txt | Txt file with commands for website in WEBSITE_code directory |
| P-racker Video | Link | https://youtu.be/dJ0f-SAdiM0 (accessed on 30 April 2025) |

**Author Contributions:** M.J. was responsible for conceptualization, data curation, formal analysis, investigation, methodology, software, validation, visualization, writing the original draft and review and editing. H.Q. was responsible for formal analysis, investigation, methodology, writing—review and editing. M.L. was responsible for conceptualization, funding acquisition, methodology, project administration, writing—review and editing. D.O. was responsible for conceptualization, funding acquisition, methodology, project administration, writing—review and editing. S.M.C. was responsible for conceptualization, data curation, formal analysis, funding acquisition, methodology, project

administration, supervision, writing—review and editing. All authors have read and agreed to the published version of the manuscript.

**Funding:** This work was supported by Taighde Éireann—Research Ireland under the following grant number: the Insight Centre for Data Analytics (12/RC/2289_P2) and the National Challenge Fund (22/NCF/HE/11177G).

**Institutional Review Board Statement:** Not Applicable.

**Informed Consent Statement:** Not Applicable.

**Data Availability Statement:** The original contributions presented in this study are included in the article/Supplementary Material. Further inquiries can be directed to the corresponding authors.

**Conflicts of Interest:** The authors declare no conflicts of interest.

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
