# Peer review of "P-Tracker: Design and Development of a Low-Cost PM2.5 Monitor for Citizen Measurements of Air Pollution"

_2813-6640, doi:10.3390/hardware3040012_

Round 1

Reviewer 1 Report

Comments and Suggestions for Authors

This article is devoted to the development of a special device for determining air quality. These studies are quite important, as air quality indicators have a significant impact on human health and the quality of life in general. This study describes the technology of creation, software features and instructions for connecting the device (which the authors called the P-Tracker) to a computer and instructions for assembling this device. Practical tests of the effectiveness of this device have been conducted, consisting in a sufficient number of tests conducted both outdoors and indoors, which also indicates a complete and detailed study of the device's operation.

All sections of the article are described in some detail, although there are comments on some of them listed below.:

  1. The authors need to talk in more detail about their work experience in this industry, if they have this experience.
  2. The cost of the device is not calculated accurately enough. The cost of the P-Tracker device is calculated by components, and the cost of other devices is calculated by market price. It is necessary to provide a more consistent comparison of the cost of the device listed by the authors in the manuscript with other air quality assessment devices.
  3. It is necessary to check the numbering of the figures, the article contains two figures with the number 1 and the missing number 3.
  4. Line 75 contains articles that talk about devices that do not have Wi-Fi and Bluetooth connections, but Section 5.1 shows devices that have all types of connections for comparison. There is no need to refer in the introduction to articles about devices that have more modern analogues.

Reviewer 2 Report

Comments and Suggestions for Authors

The manuscript presents the P-Tracker, a custom-built, low-cost, and portable PM2.5 monitoring device utilizing the SDS011 sensor and open-source hardware/software components. The work is commendable for its goal of empowering community-based environmental monitoring through an open-access, GPS-integrated, and cloud-connected platform. The design is thoughtfully validated across four distinct environments—controlled combustion, outdoor, indoor, and kitchen—demonstrating robust performance and alignment with the mission of Hardware.

While the device is well-conceived and the manuscript is clearly written and logically structured, the novelty is somewhat limited due to the widespread use of SDS011-based monitors in previous open hardware efforts (e.g., Luftdaten, AirGradient). The contribution would be strengthened by explicitly comparing P-Tracker with these existing platforms and highlighting unique advantages in design, usability, or data infrastructure.

To enhance the manuscript’s clarity and practical value, the following suggestions are offered:

  1. Bill of Materials (BOM) and Cost Transparency: Include a complete BOM and a detailed cost breakdown directly in the main text, not only in the supplementary materials.

  2. Calibration Methodology: Clarify whether the sensor was calibrated in the field or relies solely on factory calibration. If reference-grade instruments were used for validation, please specify.

  3. Sensor Limitations and Maintenance: Discuss the SDS011’s known limitations, including humidity sensitivity, limited response speed under rapidly changing conditions, and its operational lifespan (approximately 8,000 hours). Consider outlining potential failure modes and offering maintenance or replacement guidance.

  4. Lifespan-Cost Consideration: Provide an estimation of long-term operating cost. For instance, assuming 8 hours of daily use, the SDS011’s lifespan would translate to ~2.7 years. At USD 25 per sensor, the annualized cost would be approximately USD 9.30. Including this in the discussion would help contextualize the device’s sustainability and cost-effectiveness.

  5. Comparison with Related Projects: Strengthen the discussion by referencing additional community-led open hardware initiatives (e.g., PurpleAir, Sensor.Community). Positioning the P-Tracker within this landscape will help readers better understand its unique contributions.

  6. Modularity and Future Enhancements: Consider discussing whether the design allows for modular sensor replacement or integration of additional sensing capabilities (e.g., VOCs, CO2), which would increase the system’s adaptability and relevance.

The manuscript has strong potential as a reproducible and socially impactful contribution to open environmental sensing. Addressing the above aspects will further enhance its scientific value and practical applicability.
